# Exposure to air pollutants contributes to increased rate of neovascular age-related macular degeneration in Israel

**Alon Sela**[1]*, **Rinat Levinshtein**[2], **Shiri Shulman**[3,4]*

1 Agricultural and Biosystems Engineering, Agriculture Research Organization (ARO), Rishon LeZion, Israel, 2 Department of Industrial Engineering, Tel Aviv University, Tel Aviv, Israel, 3 Assuta Medical Centre, Tel-Aviv, Israel, 4 Faculty of Health Sciences, Ben-Gurion University of the Negev, Beer Sheva, Israel

* alonsela@volcani.agri.gov.il (AS); shirishu@assuta.co.il (SS)

## Abstract

Age-related macular degeneration (AMD) is a multi-factorial degenerative disease of the retina and the leading cause for vision loss in the developed world. Air pollution is considered the greatest environmental threat to public health globally. Accumulating evidence indicates that air pollution may be a modifiable risk factor for chronic eye diseases of the lens and retina, including AMD. We examined the concentration of seven air pollution particles and their influence on the prevalence of neovascular AMD in Israel. Records of patients with AMD between 2016 and 2019 were crossed with their residential areas and correlated with pollution data. AMD rates were correlated with 5 types of gas: nitrogen dioxide ($NO_2$), nitrogen oxide (NO), carbon monoxide (CO), ozone ($O_3$), sulphur dioxide ($SO_2$), and particulate matter - $PM_{2.5}$ and $PM_{10}$. A total of 93 localities across Israel were included in the analysis. AMD rates were higher in localities with greater air pollution. $NO_2$, $NO_x$, and $PM_{2.5}$ were positively correlated with AMD rates, while $O_3$ was negatively correlated with AMD rates. However, analysis of the effect of all air pollutant particles combined, showed a complex and highly non-linear effect on AMD rate, with the strongest non-linearity observed for carbon monoxide. NO2, NOx, and PM2.5 contribute to higher rate of AMD in Israel while $O_3$ seems to have a protective role (probably due to ultraviolet filtering) on AMD rates. The interaction between air pollutants and AMD seems to be complex and non-linear and should be further studied.

## Introduction

Air pollution is considered the greatest environmental threat to public health around the world. Approximately 7 million people die prematurely every year due to air pollution [1]. Air pollutants are classified as primary or secondary. Primary pollutants are directly emitted to the atmosphere and include particulate matter (PM), black carbon, sulphur oxides ($SO_2$), nitrogen oxides ($NO_x$), ammonia ($NH_3$), carbon monoxide (CO), methane, non-methane volatile organic compounds, including benzene, and certain metals and polycyclic aromatic hydrocarbons, including benzo[a]pyrene. Secondary pollutants are formed in the atmosphere

**Data availability statement:** Data and code are available from https://osf.io/zvtae/ The medical data is available only on an aggregative level (city level) due to patient care privacy. The code and the aggregated data is available in a designated Open Science Foundation repository.

**Funding:** Ariel-Assuta mutual research grant ARC2. The funders had no role in study design, data collection and analysis, decision to publish, or preparation of the manuscript.

**Competing interests:** The authors have declared that no competing interests exist.

from precursor gases through chemical reactions and microphysical processes, and include PM, ozone ($O_3$), nitrogen dioxide ($NO_2$) and several oxidised volatile organic compounds [2].

Air pollution sources are both natural and man-made. Man made air pollution originates in industry, diesel and petrol engines, friction from brakes and tyres, building and construction dust and road surfaces. Natural sources of air pollutants include volcanoes, sea spray, pollen and soil [2,3]. PM is classified by its size. $PM_{2.5}$ are particles < 2.5/1000 micron in diameter. Similarly, $PM_{10}$ refers to larger particles with a diameter < 10 microns. Some particles, such as dust, dirt, or smoke, are relatively large or dark enough to be seen with the naked eye.

Due to their ability to penetrate the blood stream through the lungs, small-sized particles of 2.5 micron/1000 mm or lower ($PM_{2.5}$), have been shown to be more harmful than larger sized particles. Specifically, they have been associated with increased risk for cardiovascular disease, including stroke [4–6], hypertension [7,8], oxidative stress [9], various malignancies, and chronic airway inflammatory diseases [10]. Accumulating evidence indicates that air pollution may be a modifiable risk factor for chronic eye diseases of the lens and retina [11].

Age-related macular degeneration (AMD) is a multi-factorial degenerative disease of the retina that manifests in degeneration of the retinal pigment epithelium and choriocapillaris via several pathways including oxidative damage and complement activation [12]. Late AMD may be associated with abnormal blood vessel growth under the retina which cause fluid accumulation, haemorrhages and eventually scarring with eventual visual loss. In such neovascular AMD cases, treatment with intravitreal injections of anti-vascular endothelial growth factor (anti-VEGF) drugs is indicated. This disease, and specifically its neovascular form, is a leading cause of vision loss in the developed world. It is estimated that in 2020, AMD caused moderate or severe vision loss in 6.22 million people and blindness in 1.84 million people [13]. It has been estimated that by 2050 the number of new cases of early and late AMD would reach 39.05 million and 6.41 million, respectively [14].

The aetiology of AMD is not clear. While age is the most consistent risk factor, additional risk factors may play a role, including ethnicity, genetics, oxidative stress, hypertension, lifestyle habits (diet, smoking) and environmental factors [15,16].

The effect of various air pollutants, on the risk for developing AMD has been studied in several countries, including Taiwan [17] where $PM_{2.5}$ particles were associated with AMD, while other studies [18] studied this association for CO and $NO_2$ particles. Dust storms was associated with AMD in Taiwan [19], and $PM_{2.5}$ particles in China [20]. $PM_{2.5}$, $PM_{10}$, $NO_2$ and CO in Korea [21] and in Canada $PM_{2.5}$ [22]. In the UK, $PM_{2.5}$ was associated with self-reported AMD while PM10, NO2 and NOx affected retinal thickness abnormalities [23]. Most of these studies investigated the relationship between PM2.5 and AMD; but the relationship between all specific air pollutants and AMD in one single place is still lacking. We aimed here to further solidify the relationship between AMD and 7 different air pollution types (including Ozone), based on data from 77 localities in Israel to better understand the complex interactions between air pollution and the development of AMD. All Data and code are fully available on the project repository (Supporting data and code are found in the Open Science Foundation (OSF) page (https://osf.io/zvtae/).

## Methods

This retrospective analysis was approved by the Assuta Medical Centres' Ethics Committee (approval number ASMC-0001-19). The need for patient informed consent was waived due to its retrospective nature.

### Air pollution data

Records related to yearly air pollution data between 2016 and 2019 were obtained from the Israel Ministry for the Protection of the Environment.

Age distributions for localities across Israel in 2017 were obtained from the Israel Central Bureau of Statistics. This dataset included statistical data related to ethnicity distributions and locality size stratified by the number of residents (10,000–20,000, 20,000–50,000, 50,000–100,000, 100,000–200,000, 200,000–500,000, over 500,000).

## Age-related macular degeneration patient data

Records of patients with neovascular AMD who received intravitreal anti-VEGF injections were retrieved from Assuta Medical Centres' Eye Clinic, where members of Maccabi Healthcare Services are treated.

Maccabi Healthcare Services is the second largest health maintenance organization in Israel (one of the four health maintenance organizations in Israel, which every citizen and permanent resident is entitled to be insured in under the National Health Insurance Law 1994). The initial database consisted of 90,000 records of patient visits to Assuta Eye Clinic between January 2016 and December 2018. Each record included open text describing the diagnostic procedure of an eye examination or an anti-VEGF injection that was given to a patient. The patient records were first anonymised and then grouped by the patient's city of residence. For this initial preprocessing the data was accessed between June 2020 to August 2020. This preprocessing resulted in an aggregated number of AMD patients in 184 locations in Israel. Overall, the data included 3200 patients.

This dataset was combined with another dataset from the census of the Israel Central Bureau of Statistics[24] and the proportion of members in each health maintenance organization per city was determined. Records of small localities, in which the total number of Maccabi Healthcare Services members was less than 300 were excluded from the analysis, as well as localities that had less than 20 patients with AMD. The final analysis included 93 localities.

The localities were also separated into mostly Hebrew speaking, mixed Hebrew and mostly Arabic speaking localities, from which some limited reference to ethnicity can be made. One should note however that this distinction should be taken with great care, since in the population of mostly Hebrew speaking localities, about 60% of Hebrew speaking Maccabi pacients come from Arab countries and in terms of population genetics, this population is more like African/ Middle eastern populations.

Since the proportions of healthcare maintenance organization members differs by locality, to estimate the rate of AMD in each locality, the number of individuals with AMD were divided by the proportion of Maccabi Healthcare Services members (% of Maccabi patients in the locality * locality's population).

One should note however that this data was constructed based on textual medical records of patients in the Assuta eye Clinic. The treating ophthalmologists do not collect lifestyle records such as BMI or other health related parameters. Also, the initial approval to use the data did not include an extraction of parameters such as smoking habits not weight.

This limits this retrospective study to use the data for the purpose declared and not to other purposes.

## Analysis

The association between AMD rates and air pollution was performed on seven different air pollution particles, i.e., 5 types of gas: nitrogen dioxide ($NO_2$), nitrogen oxide (NO), carbon monoxide (CO), ozone ($O_3$), sulphur dioxide ($SO_2$), and 2 particulate matter sizes: $PM_{2.5}$ and $PM_{10}$.

The influence of these air pollution particles on AMD rates in different localities was analysed by several regression models as well as by a cross correlation analysis - a method used in

nonlinear systems to detect interactions. The dependent variable in each regression model was the number of patients with AMD per 100,000 persons.

First, separate regression models were performed to examine the effect of each air pollution particle separately on the rate of AMD. Then, several multiple regression analyses models were performed to examine the effect of combinations of different air pollution particle types on AMD rate. Pearson correlation was also used to explain the correlation among different air pollution particles.

The cross-correlation analysis was used to refine and revalidate the finding of the multiple regressions. Mainly, these methods were needed since the effects between the air pollution particles and AMD seem to be non-linear. The cross-corelation method is commonly used in physics and in signal processing or in bioinformatics, where it is applied to study non-linear time series between seemingly disconnected variables [25,26]. The correlation between AMD rates and air pollution concentrations was examined while comparing these corelations to the internal "noise" that exists in the data. Although cross-correlation methods are usually used to find correlations between different points in time, here, we used these methods to determine correlations between different points in space (i.e., localities). The cross-correlation implied shuffling the data of different localities and comparing the maximal effects in the shuffled data to the non-shuffled data. Thus, the noise to signal was estimated by comparing the correlation between AMD rates and air pollutants in any city $i$, to the correlations when AMD rates were for a city $i$ but the air pollution records were for a city $i + k$ ( $k \neq 0$ ). This process was repeated for different values of k. By this method, the signal when $k = 0$ was extracted out of the noise ( $k \neq 0$ ). If the maximal correlation function appeared when the time lag was 0, and if it was significant (>2–3 standard deviations above the noise), then the signal of correlation was considered significant over the noise.

## Results

Neovascular AMD rate ranged from 0.006% to 1.04%. The rate was higher in localities in central Israel compared to localities in the periphery of the country (Fig 1a) and corresponded with higher air pollution measured in localities in central Israel (Fig 1b).

The average concentration of the 5 gases and particulate matter measured in the 93 localities are shown in Table 1. On average, CO particles had the highest concentration (166.4 ± 43.2 µg/m³) followed by $O_3$ (72.2 ± 8.7 µg/m³).

Correlation analysis showed that only $O_3$ was negatively correlated with the other air pollutants. All correlations between the other air pollutants were positive (Table 2).

The change in coefficient signs between the independent and multiple regression models (for example, the $NO_2$ coefficient changed from 3.4 in the independent [single factor] regression model to −26.7 in the multiple regression model) suggested a complex non-linear relationship between air pollution particles and the rate of AMD.

To further validate these finding, we examined the non-linear effects by additional methods. To that end, we used a cross-correlation analysis (Fig 2) which examines the associations between the seven air pollution particles, and AMD rate and compared them to the internal noise in the data (Fig 3). For this, the index of the localities was changed by a lag, such that the correlation between the pollution data of city i is examined with the AMD rates of city j. A peek correlation should be observed in Lag = 0 (x-axis), indicating a maximal correlation (i.e., above the noise in the data) only when the pairs "AMD in city i" and "air pollution in city j" are aligned with each other, i.e., (i=j). In these cross-correlation plots, the green horizontal lines show the three standard deviations limit (99% confidence), while the blue dashed line shows the two standard deviations limit (95% confidence). The correlation at lag=0 surpassed the three standard deviations limit for $NO_2$, $NO_x$, and CO pollutants, and barely for $PM_{2.5}$. The

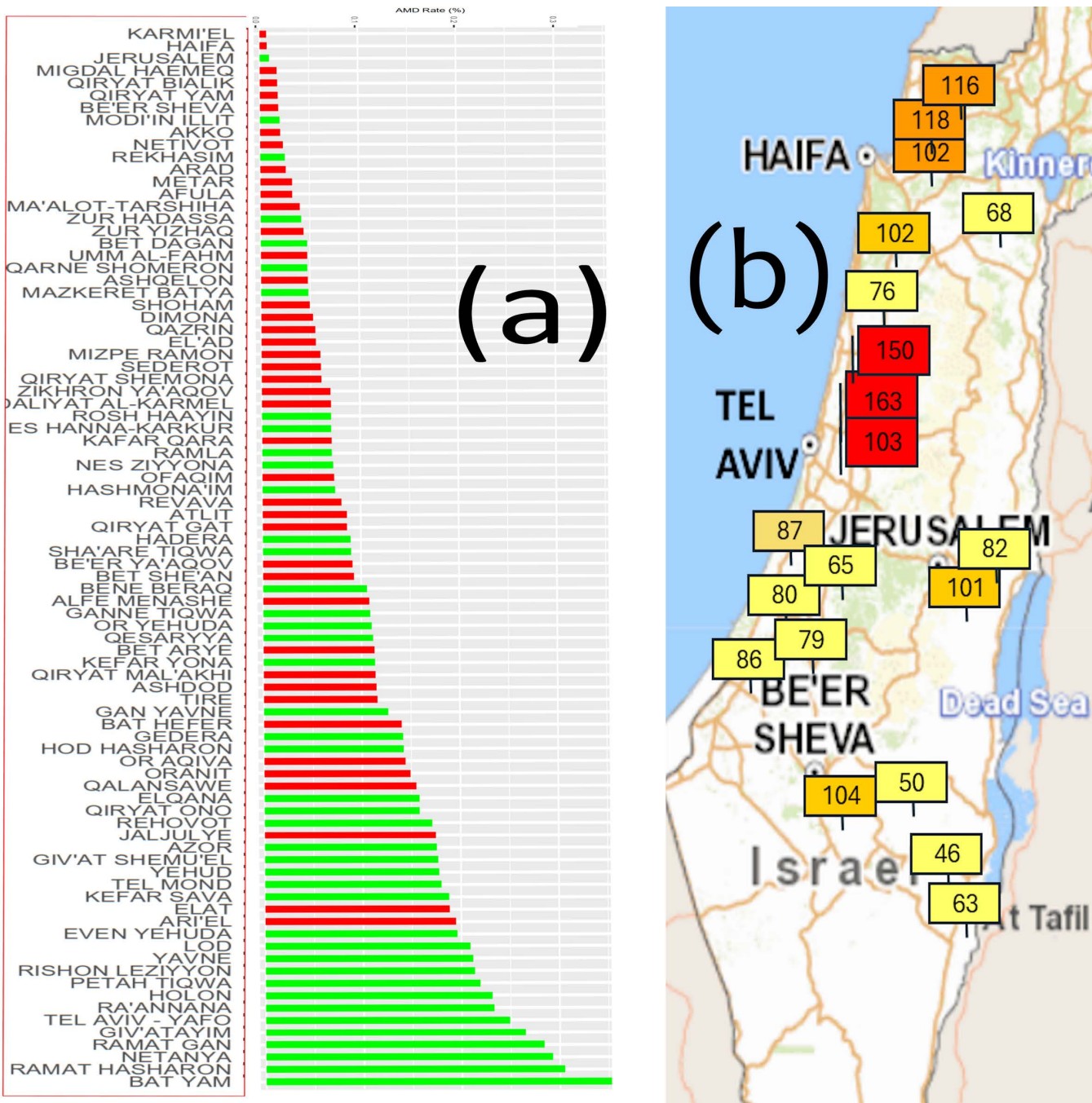

**Fig 1. AMD rate and air pollution in 93 localities across Israel.** (a) AMD rates in the 93 localities analysed. Green bars indicate localities in central Israel, red bars indicate localities in peripheral regions of the country. (b) Air pollution rates in Israel on 7 February 2022 [27]. Higher air pollution (red flags) was observed in localities in central Israel.

data shows that $PM_{10}$ surpassed the two standard deviations limits (the blue dashed line) in both directions, similar to the case of O3, in which the autocorrelation is in the negative direction. These values should be compared to the shuffled vectors, which indicate the noise. These results show a clear effect of $NO_2$, $NO_x$ CO on AMD rates and of PM2.5 to a lower extent, a weak negative effect of $O_3$, and a weaker positive effect of $PM_{10}$ and no effect of $SO_2$ particles.

**Table 1. Concentration of air pollution particles in 93 localities.**

| Particle | Concentration (µg/m³) |
|----------|------------------------|
| | **Mean ± standard deviation (Interquartile range)** |
| **CO** | 166.4 ± 43.2 (136.8–186.9) |
| **$O_3$** | 72.2 ± 8.7 (66.1–77.6) |
| **$No_x$** | 16.4 ± 9.6 (9.3–21) |
| **$NO_2$** | 12.8 ± 7 (7.2–17.2) |
| **$SO_2$** | 1.1 ± 1.1 (0.2–1.5) |
| **$PM_{10}$** | 39.5 ± 2.5 (37.8–41.3) |
| **$PM_{2.5}$** | 18.6 ± 1.8 (17.3–19.8) |

**Table 2. Pearson correlation between air pollution particles.**

| Air pollution particles | $NO_2$ | NOx | CO | $O_3$ | $SO_2$ | $PM_{2.5}$ | $PM_{10}$ |
|---|---|---|---|---|---|---|---|
| **$NO_2$** | 1 | | | | | | |
| **NOx** | 0.99 | 1 | | | | | |
| **CO** | 0.874 | 0.891 | 1 | | | | |
| **$O_3$** | −0.883 | −0.866 | −0.774 | 1 | | | |
| **$SO_2$** | 0.504 | 0.48 | 0.348 | −0.510 | 1 | | |
| **$PM_{2.5}$** | 0.803 | 0.799 | 0.697 | −0.856 | 0.459 | 1 | |
| **$PM_{10}$** | 0.645 | 0.646 | 0.639 | −0.708 | 0.463 | 0.772 | 1 |

To further demonstrate this method, the lower right and lower middle sub-images of Fig 2 are examples for $NO_2$ (which is significant), and for $SO_2$ (which is insignificant) showing the distribution of correlation values across 500 random permutations of cities. In these images, the red line represents the *actual* corelation, while the dashed green line represents the 3 standard deviations control limits. These images show that the actual correlation (red horizontal line) of $NO_2$ is far to the right compared to the 99% (3 standard deviations) control limit, while this is not so for $SO_2$.

These results strengthen our claim that *NO2*, *NOx*, *CO*, and $PM_{2.5}$ contribute to the prevalence of AMD, even if these effects might be non-linear and with complex interactions between the different gases and particles.

To conclude, in these separate analyses, the coefficients of the air pollution gases in the univariate regression were all positive, reflecting an increase in the number of neovascular age-related macular degeneration (3.4 for $NO_2$, 4.18 for $NO_x$, 0.8 for CO, 0.74 for $SO_2$) except for ozone (−3.018 for $O_3$). The effect of $SO_2$ on AMD rate was not significant in the regression analysis or in the cross-correlation method. This suggests that while greater air pollution will result in higher rates of AMD, higher rates of $O_3$ are likely to result in lower AMD rates. Regarding PM, the coefficient of $PM_{10}$ was found to be lower than those of the smaller particles (14.93 and 9.89 for $PM_{2.5}$ and $PM_{10}$ respectively). In the cross-correlation analysis, the effect of the larger particles was less significant but still above the 95% confidence limits (Table 3).

Analysis of the combined effect of air pollutants showed a more complex and non-linear effect on AMD rates (Table 4). A strong non-linear effect of air pollution variables was observed from comparing the simple regression $y = f(x1)$, $y = f(x2),...,y = f(x7)$ to the multiple regression $y = f(x1, x2,..., x7)$. While in the separate regression modes all coefficients except $O_3$ were positive, and all pollutants but $SO_2$ were significant, in a multivariate regression model that included all 7 air pollutants the coefficients of $O_3$, $NO_2$ and $PM_{2.5}$ were

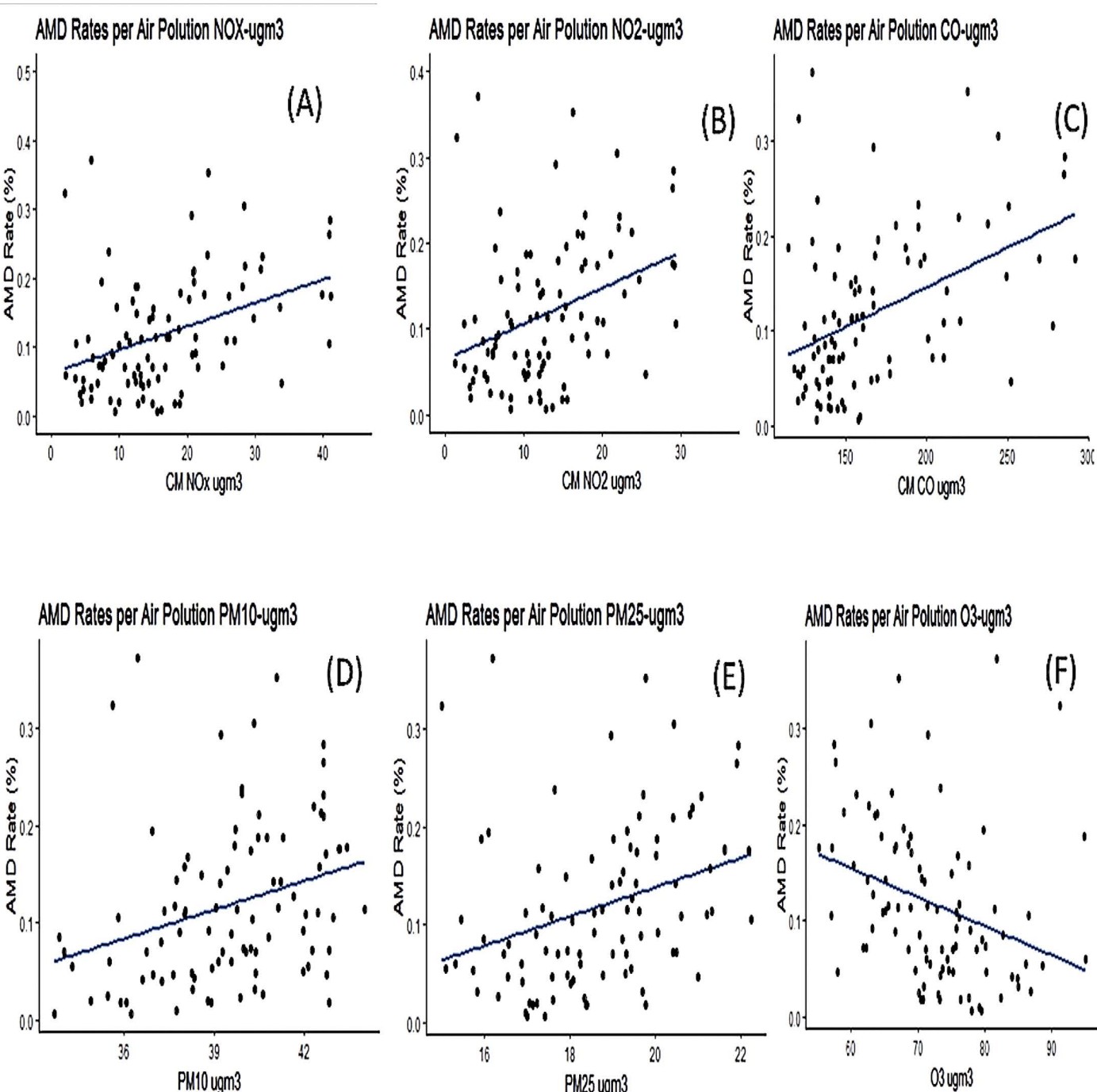

**Fig 2. Correlation between air pollutants and AMD rates (%).** Each point represents a locality. (A) NOx, $R^2 = 0.151$, $p = 0.0001$, (B) NO$_2$, $R^2 = 0.121$, $p = 0.00064$, (C) CO, $R^2 = 0.178$, $p = 1.498e{-}05$, (D) PM$_{10}$, $R^2 = 0.088$, $p = 0.0037$, (E) PM$_{2.5}$, $R^2 = 0.092$, $p = 0.00064$ (F) O3, $R^2 = 0.121$, $p = 0.0018$ For SO2 (not shown), $p = 0.926$.

negative (Table 4). The inverse results also appeared in a second multiple regression model (Table 4) that included only NO$_2$, NO$_x$ and SO$_2$, where the coefficients were −23.80, 21.27 and −10.96, respectively. These inverse coefficients can appear when the interactions between the dependent and independent variables are highly non-linear.

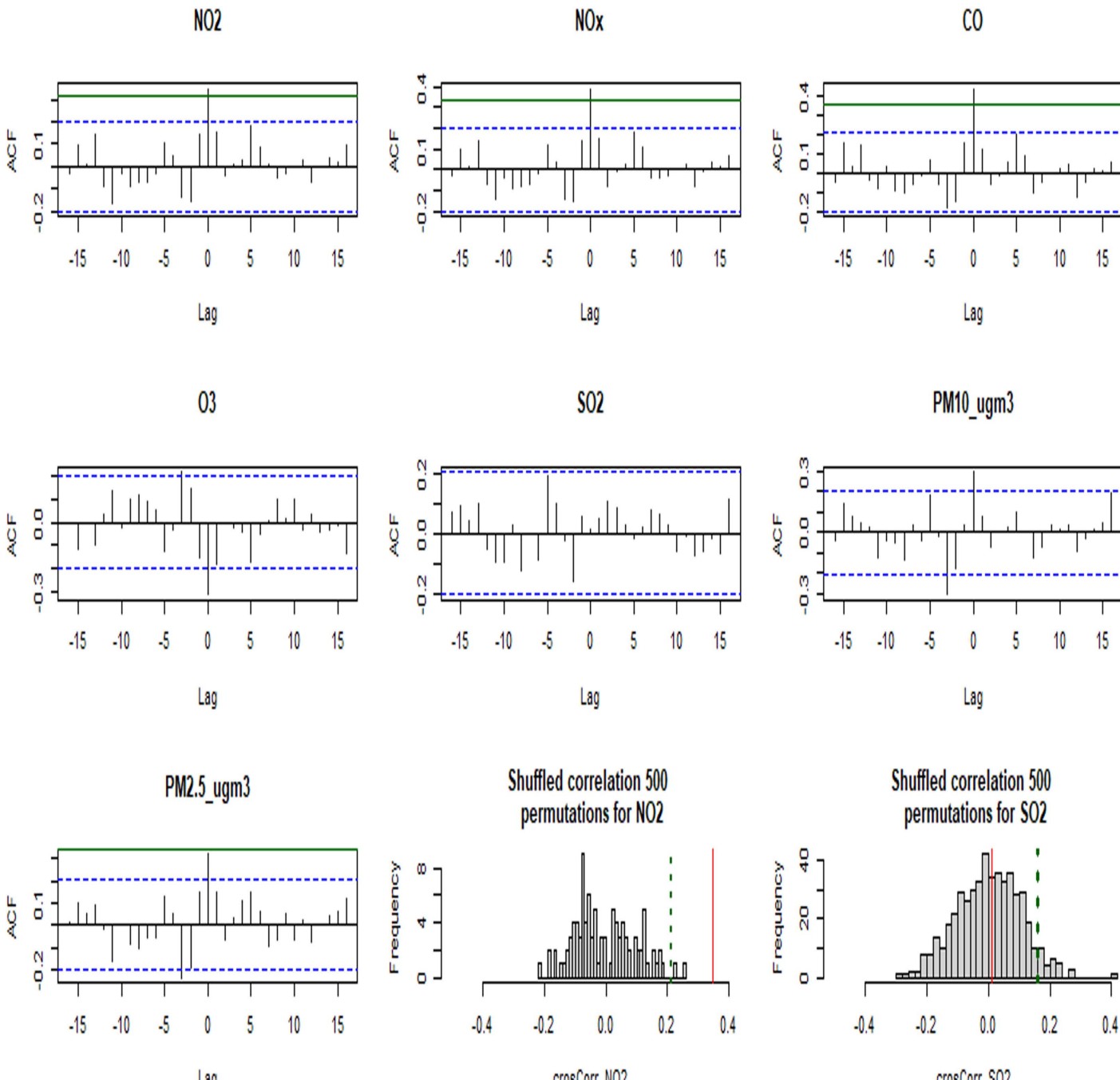

**Fig 3. Cross-correlation between air pollutant types and AMD in different localities.** If an effect is found, we expect to see the cross-correlation bar only in 0. The blue dashed line represents the statistical significance of 95%. The green line represents a statistical significance of 3 standard deviations. A clear effect of 99% (bar is above/below the green line) was observed for *NO2*, *NOx*, *CO*, and $PM_{2.5}$ particles. For $PM_{10}$ the effect was only found for 95% confidence, while the effect of *O2* and *SO2* was on the edge of acceptance, and thus might need further validation in a larger data set. The histograms are examples of the correlation values, to demonstrate the noise to signal ratio.

## Discussion

Our analysis of AMD rates and air pollution in 93 localities across Israel showed that AMD rates were higher in localities with greater air pollution. The interaction between air pollution

**Table 3. Univariate regression between AMD rate/100,000 persons and air pollution particles.**

|  | Air pollution particles in 93 localities | | | | | | |
|---|---|---|---|---|---|---|---|
|  | $NO_2$ | NOx | CO | $O_3$ | $SO_2$ | $PM_{2.5}$ | $PM_{10}$ |
| **Regression coefficients (SD)** | 3.41 (0.85)*** | 4.18 (1.18)*** | 0.837 (0.183)*** | −3.02 (0.96)*** | 0.741 (7.98) | 9.89 (3.33)*** | 14.93 (4.64)*** |
| **Constant** | 61.86 (16.03)*** | 64.19 (17.20)*** | −21.58 (31.44) | 335.66 (69.82)*** | 116.92 (12.13)*** | −272.82 (131.52)** | −160.64 (86.93)* |
| **$R^2$** | 0.151 | 0.121 | 0.187 | 0.098 | 0.0001 | 0.089 | 0.102 |
| **Adjusted $R^2$** | 0.142 | 0.111 | 0.178 | 0.088 | −0.011 | 0.079 | 0.092 |
| **Residual standard error (df = 91)** | 77.49 | 78.86 | 75.84 | 79.88 | 84.11 | 80.29 | 79.70 |
| **F statistic (df = 1;91)** | 16.22*** | 12.51*** | 20.93*** | 9.89*** | 0.009 | 8.85*** | 10.35*** |

*$p < 0.1$. **$p < 0.05$. ***$p < 0.01$.

**Table 4. Multivariate regression between AMD rate/100,000 persons and air pollution particles.**

| Regression Step | Regression coefficients (standard deviation) | | | | | | | Constant | $R^2$ | Adjusted $R^2$ | Residual standard error | F statistic |
|---|---|---|---|---|---|---|---|---|---|---|---|---|
|  | $NO_2$ | NOx | CO | $O_3$ | $SO_2$ | $PM_{2.5}$ | $PM_{10}$ |  |  |  |  |  |
| **(1)** | −26.73 (10.95)** | 21.37 (8.37)** | 0.18 (0.61) | −2.60 (2.91) | −11.01 (9.45) | −12.22 (15.46) | 6.79 (5.23) | 239.23 | 0.27 | 0.21 | 74.02 (df = 85) | 4.64*** (df = 7;85) p = 0.0002 |
| **(2)** | −23.80 (9.41)** | 21.27 (6.66)*** | – | – | −10.96 (8.83) | – | – | 85.61*** | 0.25 | 0.22 | 73.44 (df = 89) | 10.12*** (df = 3;89) p < 0.0001 |

*$p < 0.1$. **$p < 0.05$. ***$p < 0.01$.

particles and AMD seems to be complex and non-linear. All particles examined, except $O_3$, were positively correlated with higher AMD rates. $O_3$ seems to have a protective role (probably due to ultraviolet [UV] filtering) on AMD rates. However, analysis of the effect of all air pollutant particles combined showed a complex and non-linear effect on AMD rate.

One possibility that needs to be overruled is that this is simply due to older people preferring larger cities (which are more polluted generally). To check this possibility, we calculated first the mean ages in each city. We then plot these cities mean ages against the major air pollution particles.

We could not find any evidence that cities with higher mean population ages are more polluted cities. We also validated this issue statistically. We separated the localities into two groups: One consisting of cities with an above average age and the second with those with below average ages. We then compare (t-test two sided) the mean levels of air pollution particles between the two groups. The p-values of these 7 t-tests are: 1. p-values = 0.63 (No2), 2. p-values = 0.55 (NOx), 3. p-values = 0.29 (CO), p-values = 0.90 (O3), p-values = 0.68 PM10, p-values = 0.38 (PM2.5), p-values = 0.12 (So2). As a p-value < 0.05 reflects a significant difference, based on these tests, we reject the hypothesis that cities with older citizens also have higher levels of air pollution. We conclude therefore that air pollution is likely to be the cause of the higher prevalence of AMD, and in any case, this is not due to polluted localities consisting of older habitants.

One needs to notice that AMD and air pollution were collected between 2015–2020 (AMD)/ 2016–2019 (Air pollution). This period does not permit a true longitude study of air pollution over many rears. Nevertheless, since air pollution is also highly correlated with the type of the location (large city blocks, small villages in rural regions or industrial zones), changes in which rural regions become highly industrial require tens of years.

Future studies should inspect longer periods, and possibly, check for the AMD rates in regions which were once rural, and later became highly industrials. It might be challenging however to find such rejoins since industrial rejoins tend to grow at the peripheral sides of cities and less in fully rural places.

Most studies published to date have examined the relationship between exposure to $PM_{2.5}$ and the risk for AMD. In a population-based cohort study conducted in Taiwan (n = 4,284,128) between 2001 and 2011, the annual mean $PM_{2.5}$ exposure during the study period was 34.23 ± 7.17 $\mu g/m^3$, and a chronic exposure to $PM_{2.5}$ increased the risk of AMD by 19% (95% confidence interval [CI] 1.13–1.25) for each 10 $\mu g/m^3$ increase in $PM_{2.5}$. However, the risk reached a plateau and even showed some decreasing trend when $PM_{2.5}$ was higher than 35 $\mu g/m^3$, suggested to be due to a ceiling effect or competing lethal diseases [17]. Similarly, a study with participants of the UK Biobank (n = 115,954) reported an increased risk for self-reported AMD with continuous increasing concentrations ($\mu g/m^3$) of $PM_{2.5}$ (odds ratio [OR] = 1.08 [95% CI 1.01–1.16], p = 0.036), thinner photoreceptor synaptic region, thicker photoreceptor inner segment layer and thinner retinal pigment epithelium [23]. A cross-sectional analysis using data from the Canadian Longitudinal Study on Aging (n = 30,097), reported an increased risk of self-reported visually impairing AMD with an increase of one interquartile range (IQR) in $PM_{2.5}$ (OR = 1.52 [95% CI 1.10–2.09]); however, in multi-pollutant models after adjustment for sociodemographic characteristics and disease, increased $PM_{2.5}$ was not found to be associated with AMD. Furthermore, the mean $PM_{2.5}$ level was 6.5 $\mu g/m^3$ [22] – lower than the mean value reported in the study conducted in Taiwan [17]. In a national cross-sectional survey conducted across 10 provinces in rural China (n = 36,081), the average annual $PM_{2.5}$ level during the study period was 63.1 ± 15.3 $\mu g/m^3$. A significant positive association was detected between AMD and $PM_{2.5}$ level, temperature, and relative humidity, in both independent and combined effect models [20].

In a study that investigated the association between ambient air pollution and AMD in 15,115 middle-aged and older adults (≥40 years) from the Korean National Health and Nutrition Examination Survey 2008–2012, ambient $NO_2$ and CO in current-to-5 prior years and $PM_{10}$ in 2-to-5 prior years were significantly associated with higher prevalence of early AMD, while $O_3$ in current-to-5 prior years was significantly associated with lower prevalence of early AMD. Interestingly, the relationship between air pollution and the prevalence of early AMD was more significant than the relationship between air pollution and the prevalence of late AMD, suggesting that air pollution affects the early stages of disease development [21]. A prospective analysis using Taiwan's Longitudinal Health Insurance Program between 2000 and 2010 (n = 39,819) showed a higher risk for AMD in those exposed to high versus low concentrations of $NO_2$ (hazard ratio [HR] = 1.91 [95% CI 1.64–2.23], p < 0.001) and CO (1.84 [95% CI 1.5–2.15], p < 0.001) [18].

In another study conducted in Taiwan, AMD cases were also significantly associated with exposure to dust storm events, which carry high concentrations of PM [19].

As to these previous studies, the link between air pollution and AMD in [17], examined PM2.5 exposure using satellite-based remote sensing but ignored a known bias from sensing air pollution in upper atmospheric layers. In contrast [18], focused on ground-level traffic-related pollutants ($NO_2$ and CO), aligning with our approach, but excluded a broad set of pollutants such as $O_3$, $No_x$, $SO_2$, $PM_{10}$ and $PM_{2.5}$. Study [19] investigated natural dust storms, emphasizing non-anthropogenic factors. The large-scale Chinese study [20], focused solely on $PM_{2.5}$, while the Korean study [21] included $O_3$, $NO_2$, CO, and PM10 but excluded $No_x$, $SO_2$ and $PM_{2.5}$. Notably, as in our case, the Korean study suggested protective effects of $O_3$ by prevention of UV exposure, but did not elaborate on UV-induced DNA damage pathways. Differences in climate and genetic diversity between Korea and Israel further distinguish these studies.

The Canadian study [22] identified $PM_{2.5}$ as the sole AMD-related pollutant, potentially due to its linear analysis method and reliance on self-reported AMD stages and ignored the other air pollutants. By comparison, the UK study [23] used (as we did) ophthalmologist-diagnosed AMD based on OCT imaging (and not self-reported questionnaires), finding $PM_{2.5}$ linked to higher self-reported AMD rates, while $No_x$, $NO_2$ and $PM_{10}$ were associated with retinal anomalies typically linked to AMD, though requiring expert analysis for confirmation.

Our study stands and contributes to the growing evidence on air pollution's detrimental effects on AMD. It analyses the full and comprehensive set of pollutant, while defining AMD based on OCT imaging rather than self-reports. Unlike remote sensing methods, the ground-level pollution measurements offer greater accuracy.

As air pollution's impacts extend beyond the known global warming effects and as we show here (and by others) include real and concrete vision impairments for older adults. With the expected growth in life expectance and the growing levels of air pollution, further attention to this issue is warranted.

Several potential mechanisms underlie the effects of air pollution on AMD. $PM_{2.5}$ has been associated with poor retinal structure, which may lead to AMD [23,27].

Exposure to high levels of PM, $SO_2$, $NO_2$, $NO_x$, CO, and $SO_2$ can cause cellular damage through systemic oxidative stress, consequent lipid peroxidation, which activates the innate immune system and increases inflammation in the retina and cells, potentially resulting in increased risk of AMD [17,21,28,29]. The non-linear effects observed between air pollution and the rate of AMD might be explained through two separate DNA repair mechanisms: one repairing damage resulting of UV radiation and the other repairing oxidation damage. Oxidative stress reflects an imbalance between the production of reactive oxygen species and the body's ability to remove it. Reactive oxygen species cause DNA base modifications as well as strand breaks. The primary mechanism for repairing oxidative damage is base excision repair, which involves the removal of the damaged DNA base and its replacement with a new one. UV damage primarily occurs through the formation of cyclobutene pyrimidine dimers and 6–4 photoproducts. These lesions block replication and transcription, leading to mutations and cell death. The primary DNA repair mechanism responsible for UV damage is the nucleotide excision repair [30]. which involves the removal of a segment of DNA containing the damaged lesion and its replacement with a new, undamaged strand. High air pollution rates seem to be associated with higher rates of ground level $O_3$, which were shown to filter the UV radiation [31]. As $O_3$ filters UV radiation it may protect the retina from damage, in line with our findings that greater exposure to $O_3$ reduced the prevalence of AMD. Furthermore, regions in the world (such as Tibet, Nepal, and other high-altitude regions), which have relatively low air pollution, but high rates of UV radiation, have increased rates of AMD [32, 33]. While these are mainly possible assumptions, the exact mechanism underlying the contribution of air pollution to AMD still requires further research.

This study's strength lies in its evaluation of the effects of several known gas and PM pollutants. Additionally, the diagnosis of AMD was according to treatment criteria, which are more accurate than self-report or other forms of electronic medical records diagnosis.

The main limitation of the study is its retrospective nature. In addition, the changes over time could not be addressed. Furthermore, there may be long time lag between exposure to air pollution and AMD diagnosis. Lastly, we only addressed rates of neovascular AMD, which is a late manifestation of AMD that may also be affected by several other factors.

## Conclusion

In the current work, we examined seven air pollution particles and their influence on the prevalence of AMD in 93 localities across Israel. NO2, NOx, CO and $PM_{2.5}$ and $PM_{2.5}$ were

positively correlated with AMD rates, while $O_3$ was negatively correlated with AMD rates. However, analysis of the effect of all air pollutant particles combined showed a complex and non-linear effect on AMD rate, where only NO2 and NOx significantly influenced the rate of AMD. When using cross-correlation methods, which better fit an analysis of nonlinear phenomena, we found that CO is also strongly correlated with AMD. The predicted growth in life expectancy and environmental pollution are expected to increase the prevalence of AMD. A growing body of literature is accumulating from different countries, ethnicities and climates, pointing to the harmful effect of air pollution and its clear influence on AMD. While reducing air pollution is a major concern to global warming, increased rates of blindness at old age needs to alert decision makers no less, especially with prolonged lifespan.

Further investment in research related to this alerting topic needs to be concluded, to better understand the exact biological paths by which air pollution damages the retina. This may also shed light on the pathogenesis of this complex degenerative disease and possibly to find additional treatments. The role of countries and health organization is to raise awareness to this additional effect of air pollution on AMD and help to further reduce the rates of this blinding disease and by this - to reduce long term treatment expenses.

## Supporting information

**Table S1. Descriptive statistics.** For initial air pollution data (N = 1,214 localities).
(DOCX)

**Fig S1. AMD rates by population age.** Rates of AMD in different localities across Israel.
(PNG)

**Fig S2. Mean ages at different localities and their air particle concentration.** (A) No2, (B), NOx, (C) CO, (D) PM2. Also, t-test were performed to check if cities with older populations have significantly more air pollution. For all seven air pollution particles, we reject this possibility.
(PNG)

**Fig S3. AMD age distribution by gender.** We find gender is rather equally distributed between the cities. Nevertheless, as seen in the image below, more women in our data suffer from AMD in age ranges of 75-90. This could also be due to different life span between genders.
(PNG)

**Fig S4. AMD rates per ethnicity and Nox concentration.** While there are restrictions on ethnicity labelling in medical records in Israel. We do however have language-based clues for ethnicities. (1) represent Jewish towns (where the spoken language is Hebrew). (2) are mixed towns (where ethnicities are mixed, and the spoken language are Arabic and Hebrew, i.e., the population in a mixture of Arabs and Jews, (3) are towns where the spoken language is mostly Arabic, and ethnicities are mostly of Arab origins. One should note that the spoken language is not a clear ethnical recognition, as over 60% of Jewish population is of middle eastern origin.
(PNG)

**Fig S5. AMD and ethnicities in the sample.** While it seems like AMD is less prevalent in Arab towns, the data set of Maccabi health care is also balanced toward Jewish towns (which are more common in Maccabi health care). Further research is required to conclude this claim on a larger sample.
(PNG)

**Fig S6. Total air pollution (normalized) and AMD cross correlation.** Stronger correlation is clearly observed at LAG = 0, when city air pollution and AMD rates are aligned. There are restrictions on a full ethnicity labelling in medical records in Israel. We do however have language-based clues for ethnicities. (1) represent Jewish towns (where ethnicities can be highly varied). (2) are Arab towns, where ethnicities are mostly middle eastern. (3) are mixed towns, where again, ethnicities are mixed. While it seems as AMD is less prevalent in Arab towns, the data set is balanced toward Jewish towns (which are more common in Maccabi health). Further research is required to conclude this claim on a larger sample.
(PNG)

## Author contributions

**Conceptualization:** Alon Sela.

**Data curation:** Alon Sela.

**Formal analysis:** Alon Sela, Rinat Levinshtein.

**Funding acquisition:** Shiri Shulman.

**Investigation:** Alon Sela.

**Methodology:** Alon Sela.

**Project administration:** Alon Sela, Shiri Shulman.

**Resources:** Shiri Shulman.

**Software:** Alon Sela.

**Supervision:** Shiri Shulman.

**Validation:** Shiri Shulman.

**Visualization:** Alon Sela.

**Writing – original draft:** Alon Sela, Rinat Levinshtein, Shiri Shulman.

**Writing – review & editing:** Alon Sela, Shiri Shulman.

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
