## [Decision Letter · Decision Letter 0]

20 Mar 2024

PONE-D-23-40090Exposure to air pollutants contributes to an increased rate of neovascular age-related macular degeneration in IsraelPLOS ONE

Dear Dr. Sela,

Thank you for submitting your manuscript to PLOS ONE. After careful consideration, we feel that it has merit but does not fully meet PLOS ONE’s publication criteria as it currently stands. Therefore, we invite you to submit a revised version of the manuscript that addresses the points raised during the review process.

We look forward to receiving your revised manuscript.

Kind regards,

Mohd Akbar Bhat, Ph.D.

Academic Editor

PLOS ONE

Journal Requirements:

Ariel-Assuta mutual research grant ARC2. 

4. In the online submission form, you indicated that Data are available from the Alon Sela alonse2012@gmail.com upon request.

The medical data is available only on an aggregative level (city level) due to patient care privacy.

The code and the aggregated data will be available in a designated GitHub repository upon a review request.

6. Please amend either the title on the online submission form (via Edit Submission) or the title in the manuscript so that they are identical.

7. We note that Figure 1 in your submission contain map images which may be copyrighted. All PLOS content is published under the Creative Commons Attribution License (CC BY 4.0), which means that the manuscript, images, and Supporting Information files will be freely available online, and any third party is permitted to access, download, copy, distribute, and use these materials in any way, even commercially, with proper attribution. For these reasons, we cannot publish previously copyrighted maps or satellite images created using proprietary data, such as Google software (Google Maps, Street View, and Earth). For more information, see our copyright guidelines: http://journals.plos.org/plosone/s/licenses-and-copyright.

We require you to either present written permission from the copyright holder to publish these figures specifically under the CC BY 4.0 license, or remove the figures from your submission:

Reviewers' comments:

Reviewer's Responses to Questions

**Comments to the Author**

1. Is the manuscript technically sound, and do the data support the conclusions?

Reviewer #1: Yes

Reviewer #2: Partly

2. Has the statistical analysis been performed appropriately and rigorously? 

Reviewer #1: Yes

Reviewer #2: I Don't Know

3. Have the authors made all data underlying the findings in their manuscript fully available?

Reviewer #1: Yes

Reviewer #2: No

4. Is the manuscript presented in an intelligible fashion and written in standard English?

Reviewer #1: Yes

Reviewer #2: Yes

5. Review Comments to the Author

Reviewer #1: AMD is a severe problem related to vision loss. There are many factors which might contribute AMD but age is the most important factor. Here in this article the author is trying to relate AMD with air pollutant. Since air pollution is on the rise and the author has shown how these air pollutant has direct relation with AMD, this kind of study is very important. However the author has not mentioned about the age of the patients. It will be important to see if there are any patient with AMD relatively at young age. It is also interesting to see if the author has any data with AMD and air pollutant in relation to the sex of the patient.

Reviewer #2: I would like to thank you for your efforts on the manuscript entitled “Exposure to air pollutants contributes to an increased rate of neovascular age-related macular degeneration in Israel".

Please see my comments below:

MAJOR COMMENTS

Comment 1

There is very limited patient information available for this study. There is no information on age, sex, smoking status, BMI etc that are major contributing factors for AMD. It is not clear if the statistical data was adjusted based on such information.

Comment 2

Lines 99-101 states that, “Records of patients with neovascular AMD who received intravitreal anti-VEGF injections were retrieved from Assuta Medical Centres’ Eye Clinic, where members of Maccabi Healthcare Services are treated”.

The above statement indicates that the patients included in the study were receiving anti-VEGF therapy. Therefore, it is important to include data/information on the treatment status from routine ocular examinations, as that will significantly influence the data. For example, what percentage of these patients were responding to the therapy and showing improvement etc.

Comment 3

This study is limited to information on area level exposure. It lacks information on individual exposure, which may be affected by personal measures/lifestyle.

Comment 4

Several studies show the correlation of different air pollutants with AMD prevalence. Since this

study aimed to better understand the interactions between air pollution and the development of AMD, it would be more relevant to study the progression of AMD with variations in air pollution (and other factors) over a longer period of time.

Comment 5

The discussion includes a comprehensive compilation of similar studies in other countries. It would be compelling to compare and contrast this study to the other studies to get an insight into the importance of ethnicity in the current analysis.

MINOR COMMENTS

Comment1

Lines 192-193:

“It also surpassed the two standard deviations limits for PM10 and in the negative direction for O3”.

The data shows that PM10 surpassed the two standard deviations limits in both directions, similar to O3. Please correct the above statement with appropriate explanation.

Comment 2

Please include shuffled correlation data for all pollutants.

Comment 3

Lines 224-226:

“To conclude, in these separate analyses, the coefficients of the air pollution gases in

the univariate regression were all positively associated with an increase in the rate of

neovascular age-related macular degeneration”.

The above statement should be rephrased to “increase in the number of neovascular age-related macular degeneration”.

6. PLOS authors have the option to publish the peer review history of their article (what does this mean? ). If published, this will include your full peer review and any attached files.

**Do you want your identity to be public for this peer review?** For information about this choice, including consent withdrawal, please see our Privacy Policy .

Reviewer #1: No

Reviewer #2: No

---

## [Author Response · Author response to Decision Letter 0]

22 May 2024

Respond to reviewer has been uploaded as a separate document to the system.

Each comment of the reviewer has been answered.

We thank the reviewers again in this occasion for the time they have spent and their comments which we feel improved our manuscript.

---

## [Decision Letter · Decision Letter 1]

14 Nov 2024

PONE-D-23-40090R1Exposure to air pollutants contributes to increased rate of neovascular age-related macular degeneration in IsraelPLOS ONE

Dear Dr. Sela,

Thank you for submitting your manuscript to PLOS ONE. After careful consideration, we feel that it has merit but does not fully meet PLOS ONE’s publication criteria as it currently stands. Therefore, we invite you to submit a revised version of the manuscript that addresses the points raised during the review process.

The manuscript has been evaluated by two reviewers, and their comments are available below.

The reviewers have raised a number of major concerns. In particular, they request a clearly outlined rationale for the study and further information on the analysis.

Could you please carefully revise the manuscript to address all comments raised?

We look forward to receiving your revised manuscript.

Kind regards,

Helen Howard

Staff Editor

PLOS ONE

Reviewers' comments:

Reviewer's Responses to Questions

**Comments to the Author**

1. If the authors have adequately addressed your comments raised in a previous round of review and you feel that this manuscript is now acceptable for publication, you may indicate that here to bypass the “Comments to the Author” section, enter your conflict of interest statement in the “Confidential to Editor” section, and submit your "Accept" recommendation.

Reviewer #2: All comments have been addressed

Reviewer #3: All comments have been addressed

2. Is the manuscript technically sound, and do the data support the conclusions?

Reviewer #2: Yes

Reviewer #3: Partly

3. Has the statistical analysis been performed appropriately and rigorously? 

Reviewer #2: Yes

Reviewer #3: I Don't Know

4. Have the authors made all data underlying the findings in their manuscript fully available?

Reviewer #2: Yes

Reviewer #3: Yes

5. Is the manuscript presented in an intelligible fashion and written in standard English?

Reviewer #2: Yes

Reviewer #3: Yes

6. Review Comments to the Author

Reviewer #2: The authors have responded to all the comments to the best of their ability and made corrections wherever needed.

Reviewer #3: This is a very important topic. Faced with the impact of air pollution on health, this article has done a very complete research explanation. However, the following points need to be explained.

Introduction

1.There have been many studies on this topic in the past, and the author should clarify what is innovative or different from previous studies.

Material and Methods

2.Is the air pollution data collected only from 2016 to 2019 sufficient to represent the air pollution exposure of the subjects in this study? Has sensitivity analysis been performed?

3.Are there corrections for personal health or disease factors? For example, diabetes, high blood pressure, frequency of eye use in work or life, UV exposure, etc.?

Results

4.Please improve the quality of figures.

Discussion

5.The same question, whether there are the same or different findings from past studies. What is innovativeness.

6.In addition to exploring the issues of air pollution and health, this study should stand on a higher vantage point and make possible suggestions for policy or health countermeasures.

7. PLOS authors have the option to publish the peer review history of their article (what does this mean? ). If published, this will include your full peer review and any attached files.

**Do you want your identity to be public for this peer review?** For information about this choice, including consent withdrawal, please see our Privacy Policy .

Reviewer #2: No

Reviewer #3: No

---

## [Author Response · Author response to Decision Letter 1]

9 Dec 2024

Thank you for the time and effort spent to review our work. While reviewer 1 and 2 accepted the paper, reviewer 3 raised some issues that were carefully corrected and improved in this revision.

---

## [Decision Letter · Decision Letter 2]

29 Dec 2024

Exposure to air pollutants contributes to increased rate of neovascular age-related macular degeneration in Israel

PONE-D-23-40090R2

Dear Dr. Hannan,

We’re pleased to inform you that your manuscript has been judged scientifically suitable for publication and will be formally accepted for publication once it meets all outstanding technical requirements.

Kind regards,

Der-Chong Tsai, MD, PhD

Academic Editor

PLOS ONE

Additional Editor Comments (optional):

All concerns raised during review process have been adequately addressed.

Reviewers' comments:

Reviewer's Responses to Questions

**Comments to the Author**

1. If the authors have adequately addressed your comments raised in a previous round of review and you feel that this manuscript is now acceptable for publication, you may indicate that here to bypass the “Comments to the Author” section, enter your conflict of interest statement in the “Confidential to Editor” section, and submit your "Accept" recommendation.

Reviewer #3: All comments have been addressed

2. Is the manuscript technically sound, and do the data support the conclusions?

Reviewer #3: Yes

3. Has the statistical analysis been performed appropriately and rigorously? 

Reviewer #3: Yes

4. Have the authors made all data underlying the findings in their manuscript fully available?

Reviewer #3: No

5. Is the manuscript presented in an intelligible fashion and written in standard English?

Reviewer #3: Yes

6. Review Comments to the Author

Reviewer #3: Thanks for your response. No further comments.

The authors make response to my comments point by point.

7. PLOS authors have the option to publish the peer review history of their article (what does this mean? ). If published, this will include your full peer review and any attached files.

**Do you want your identity to be public for this peer review?** For information about this choice, including consent withdrawal, please see our Privacy Policy .

Reviewer #3: No

---

## [Editor Report · Acceptance letter]

PONE-D-23-40090R2

PLOS ONE

Dear Dr. Sela,

I'm pleased to inform you that your manuscript has been deemed suitable for publication in PLOS ONE. Congratulations! Your manuscript is now being handed over to our production team.

Kind regards,

on behalf of

Dr. Der-Chong Tsai

Academic Editor

PLOS ONE